# Multicycle Performance of CaTiO_3_ Decorated CaO-Based CO_2_ Adsorbent Prepared by a Versatile Aerosol Assisted Self-Assembly Method

**DOI:** 10.3390/nano11123188

**Published:** 2021-11-24

**Authors:** Ren-Wei Chang, Chin-Jung Lin, Ya-Hsuan Liou

**Affiliations:** 1Department of Geosciences, National Taiwan University, Taipei 106, Taiwan; d06224005@ntu.edu.tw; 2Research Center for Future Earth, National Taiwan University, Taipei 106, Taiwan; 3Department of Environmental Engineering, National Ilan University, Yilan 260, Taiwan

**Keywords:** CO_2_ adsorbent, calcium oxide, multicycle performance

## Abstract

Calcium oxide (CaO) is a promising adsorbent to separate CO_2_ from flue gas. However, with cycling of carbonation/decarbonation at high temperature, the serious sintering problem causes its capture capacity to decrease dramatically. A CaTiO_3_-decorated CaO-based CO_2_ adsorbent was prepared by a continuous and simple aerosol-assisted self-assembly process in this work. Results indicated that CaTiO_3_ and CaO formed in the adsorbent, whereas CaO gradually showed a good crystalline structure with increased calcium loading. Owing to the high thermal stability of CaTiO_3_, it played a role in suppressing the sintering effect and maintaining repeated high-temperature carbonation and decarbonation processes. When the calcium and titanium ratio was 3, the CO_2_ capture capacity was as large as 7 mmol/g with fast kinetics. After 20 cycles under mild regeneration conditions (700 °C, N_2_), the performance of CO_2_ capture of CaTiO_3_-decorated CaO-based adsorbent nearly unchanged. Even after 10 cycles under severe regeneration conditions (920 °C, CO_2_), the performance of CO_2_ capture still remained nearly 70% compared to the first cycle. The addition of CaTiO_3_ induced good and firm CaO dispersion on its surface. Excellent kinetics and stability were evident.

## 1. Introduction

The global climate-change phenomenon has become an important concern in recent years because of excessive CO_2_ emissions, and this situation will continue because our energy supply originates mostly from fossil-fuel combustion now and in the next few years. Therefore, capturing CO_2_ from flue gas and transporting it to a suitable site for storage is a solution for CO_2_-emission reduction [1,2,3]. Calcium is an abundant element on earth, and its oxide form, CaO, can capture CO_2_ and change it to the carbonate form, CaCO_3_, via carbonation. Then the oxide form can be regenerated back by CO_2_ removal through decarbonation. Therefore, CaO-based adsorbents can be used repeatedly by combining carbonation and decarbonation processes [4,5,6,7,8]. The nontoxicity and low cost of CaO-based adsorbents also enable them to become excellent potential candidate for CO_2_-adsorption application.

However, the sintering problem that causes CO_2_-capture capacity loss dramatically after few cycles remains a challenge during operation in high-temperature environments (600–900 °C) [4,9,10,11]. The process of CO_2_ capture by calcium oxide can be divided into two stages: the surface chemical-reaction controlled stage and the inner-diffusion-controlled stage [12,13]. The former has higher capture efficiency than the latter. The sintering effect increases particle size. Larger particle sizes result in the difficult use of inner calcium oxide because the latter stage is more important. Thus, the capture performance is reduced. Introducing additives with high thermal stability into CaO-based adsorbents has been a common strategy to mitigate sintering effects and improve cyclic stability. This strategy involves adding another metal (Zr, Al, Mg, Si, etc.) precursor during CaO-based adsorbent preparation, and then high-thermal-stability additives form independently (e.g., MgO, Al_2_O_3_, SiO_2_) or react with partial calcium (e.g., Ca_9_Al_6_O_18_, CaZrO_3_) as the sorbent is obtained [13,14,15,16,17,18,19], or introducing it into high-thermal-stability porous silica solid (e.g., SBA-15, KIT-6) via the impregnation process [9,20,21]. The granules form sorbents for enhancing mechanical stability, or acoustic sound-assisted technique are also considered to reduce sorbent deactivation after several repeated uses [22,23].

The incorporation of titanium additives can introduce calcium titanium oxide (CaTiO_3_) into CaO-based adsorbents. This introduction can be expected to alleviate the CaO-based adsorbent sintering problem, owing to its thermal stability at the working temperature of CaO-based adsorbents [24,25]. Yu et al. used the precipitation and deposition method to prepare Ca/Al/Ti sorbents. The capture capacity was reduced less than 5% after 10 cycles of the capture–regeneration experiment as small amount of titania were introduced. Wu et al. prepared CaTiO_3_/nano-CaO by the hydrolysis method. The authors found that the high melting point of CaTiO_3_ can improve the cyclic stability of CaO-based adsorbents. The sorbent almost retained its capacity after 40 repeated uses.

Our scope is to prepare well sintering-resisted CaO-based CO_2_ adsorbent through an easily scaled up process. In the current work, CaTiO_3_-decorated CaO-based CO_2_ adsorbent was prepared by the aerosol-assisted self-assembly method. This method can be used to quickly and continuously prepare multicomponent metal oxide [26,27]. The properties and CO_2_-capture performance of sorbents with different calcium and titanium mole ratios were investigated.

## 2. Materials and Methods

### 2.1. Adsorbents Preparation

In the typical preparation, 11.5 mL of acetic acid (Nacalai, Nacalai Tesque, Kyoto, Japan, 99%), 5 mL of hydrochloric acid (Nacalai, Nacalai Tesque, Kyoto, Japan, 35%), 60 mL of alcohol (Nacalai, Nacalai Tesque, Kyoto, Japan, >99.5%), calcium nitrate 4-hydrate (Ca(NO)_3_ · 4H_2_O, Macron, Avantor Inc., Radnor, PA, USA), and 6 g of F127 (Sigma-Aldrich, Sigma Ltd., Saint Louis, MO, USA) were added in 17 mL of tetrabutyl orthotitanate (Ti(OC_4_H_9_)_4_, Sigma-Aldrich, Sigma Ltd., Saint Louis, MO, USA, 97%) sequentially. The mole ratio of Ca(NO)_3_ · 4H_2_O to Ti(OC_4_H_9_)_4_ was 1.2/2/3/5. The mixture was stirring until all chemicals totally dissolved. Then, the transparent mixture solution was transformed to aerosol by nozzle first, and then passed through a 400 °C furnace. The as-prepared product was collected on filter paper. The sorbent was obtained by calcination at 700 °C for 30 min with a 1 °C/min heating rate. The received sorbent was named as CaTi-x, where x represents the mole ratio of calcium to titanium.

### 2.2. Adsorption Characterization

An X-ray diffraction (XRD) measurement was completed by PANalytical X’ Pert PRO (Cu Kα λ = 0.154 nm, voltage: 45 kV, current: 40 mA). A nitrogen adsorption/desorption experiment was completed by Micromeritics ASAP 2020 (degas condition: 130 °C overnight, analysis condition: liquid nitrogen environment). Surface area was calculated from the adsorption isotherm in the relative pressure range of 0.05–0.3 using BET theory. Scanning electron microscopy (SEM) was done using a JEOL JSM-6500F.

### 2.3. CO_2_ Capture Experiments

CO_2_ capture experiments were evaluated by the gravimetric method (TA SDT-Q600). Approximately 5 mg of sorbent was used and pretreated at 700 °C under pure nitrogen gas flowing for 0.5 h. The capture environment was set at 600 °C under pure CO_2_ gas flowing for 1 h. A cyclic stability experiment was done by using the gravimetric method. Each cycle combined carbonation and regeneration processes, and a pretreatment process was added before the cycle experiment started. The capture condition was set at 600 °C under pure CO_2_ gas flowing for 5 min, and the regeneration condition was set at 700 °C under pure N_2_ gas flowing for 50 min (20 cycles) and 920 °C under pure CO_2_ gas flowing for 15 min (10 cycles).

## 3. Results and Discussion

### 3.1. Properties of Adsorbents

The SEM images of obtained sorbents with different calcium to titanium mole ratios are shown in Figure 1. Owing to fixed titanium precursor addition amounts, the larger ratio corresponded with more calcium-precursor loading. At low calcium-precursor loading (CaTi-1.2 and CaTi-2), the morphology showed a more aggregate pattern than high calcium-precursor loading (CaTi-3 and CaTi-5).

The XRD result, as shown in Figure 2, revealed that CaTi-1.2 and CaTi-2 had diffraction signals assigned as CaTiO_3_ contribution (JCPDS #220153) only. With increased calcium loading, the diffraction signals’ intensities contributed by CaTiO_3_ decreased. The CaO diffraction signal was not observed in the two sorbents because less calcium was available to form a highly crystalline structure. CaTi-3 presented diffraction signals contributed by CaTiO_3_ and CaO (JCPDS #772376). The sample with higher calcium-precursor loading sorbent, CaTi-5, presented two other diffraction signals from Ca(OH)_2_ (JCPDS #841263) compared with CaTi-3. This result indicated that only certain amounts of calcium can form high-thermal-stability oxide with titanium, and the others formed oxide itself. The highly crystalline hydroxide form demanded more calcium to participate than did the oxide form.

The nitrogen adsorption/desorption analysis results are shown in Figure 3. All four sorbents showed surface areas of 5–17 m^2^/g (Table 1). Similar value indicated that calcium-precursor loading had little influence on the change in surface area. The appearance of hysteresis loops indicated a porous structure, possibly due to nanoparticle aggregation or stacking. The pore size distribution of the sorbents in this work, CaTi-2, CaTi-3, and CaTi-5, were primarily in the range from 2 to 10 nm, whereas CaTi-1.2 was mainly in the range from 8 to 22 nm (Appendix A).

### 3.2. CO_2_ Capture Performance of Adsorbents

The CO_2_-capture performance of calcium-based sorbents is presented in Figure 4. All sorbents showed rapidly increasing capture in less than 3 min before reaching saturation. CO_2_ capture by calcium oxide involved two stages: the first was a rapidly increasing capture amount controlled by chemical reaction, and the second was slowly increasing capture amount controlled by an inner-diffusion mechanism [6,13]. However, these calcium-based sorbents were mainly observed first rapidly increasing in the capture-amount stage, while the inner-diffusion stage was unclear. This result indicated that the calcium-oxide particle size in these sorbents was very small, so the calcium oxide in inner sites could be utilized easily. Wei et al. concluded that pore size located in the range from 2 to 10 nm is beneficial for the chemical reaction controlled stage [28]. Therefore, CaTi-3 and CaTi-5 more rapidly increased than the other two sorbents, which may be related to the reason that most of their pore sizes were in the range from 2 to 10 nm. The saturated capture amount increased with increased calcium-precursor loading because a higher amount of calcium oxide was present (Table 1).

### 3.3. Multicycle Capture and Regeneration Experiment of Adsorbents

The sorbents with the highest and second highest saturated capture amounts, CaTi-3 and CaTi-5, were selected for evaluation of their multicycle stability. The results are shown in Figure 5a,b, respectively. After 20 cycles of repeated use, the capture capacity of CaTi-3 was almost unchanged, whereas that of CaTi-5 decreased by ~17% compared with that in the first cycle (Figure 5c). CaTi-5 showed capacity loss because too much calcium oxide was present in this sorbent. The carbonation behavior of CaTi-3 and CaTi-5 at the twentieth cycle were different than the first cycle (Figure 5d). The two stage carbonation behavior due to more importance of the inner-diffusion mechanism at the twentieth cycle in comparison to the first cycle carbonation reaction revealed that slight sintering occurred in both sorbents during 20 cycles of repeated use. The sintering phenomenon occurred at the CaCO_3_ phase due to its low Tammann temperature (~530 °C) [13]. Thus, the inner-diffusion controlled mechanism became important as repeated times increased. This result indicated that after 20 cycles of repeated use, calcium species particle sizes increased. Additionally, the capture capacity of CaTi-3 in the twentieth-cycle carbonation was closer to its first cycle than the CaTi-5 case at 10 min, indicating that calcium species particles suffered a greater degree of sintering in CaTi-5 than in CaTi-3.

The XRD results of CaTi-3 and CaTi-5 sorbents after 1 and 20 cycles of repeated use are presented in Figure 6a,b, respectively. After 1 cycle, CaTiO_3_ and Ca(OH)_2_ diffraction signals could be found in both sorbents. However, several CaCO_3_ diffraction signals (JCPDS 850849) clearly appeared in CaTi-5 after 20 cycles of repeated use. Given that the cycle experiment involved an initial carbonation part followed by regeneration, the CaCO_3_ species remaining after the cyclic experiment indicated that decarbonation efficiency decreased. The CaCO_3_ species could not capture CO_2_, so CaTi-5-capture capacity loss occurred after 20 cycles. Moreover, one CaCO_3_ diffraction signal with a very weak peak intensity was found in CaTi-3 after 20 cycles, indicating that cyclic stability started to decrease when the mole ratio of calcium to titanium was greater than 3. The SEM images of CaTi-3 and CaTi-5 after 20 cycles are shown in Figure 7. CaTi-5 presented a denser morphology than CaTi-3, indicating a lower carbonation performance of CaTi-5 compared to CaTi-3, which was consistent with the cyclic-experiment results.

CaTi-3 was selected for further evaluation of its multicycle stability at high temperatures under CO_2_ flowing regeneration conditions because of its superior cyclic performance (Figure 8). The results showed significant capacity loss at the third cycle, probably due to microstructure change, followed by a gradual increase in capacity, which may be related to structure optimization. After the seventh cycle, the capacity became unchanged and exhibited about a 32% decrease compared with the first cycle. The weight loss during regeneration was similar to weight gain during carbonation in each cycle. Thus, the capture capacity decreased after several times of repeated use, which may be related to low carbonation efficiency. Hence, CaO was not completely utilized at the end of carbonation. The XRD results of CaTi-3 after 1 and 10 cycles of repeated use showed diffraction signals attributed to CaTiO_3_, CaCO_3_, and CaO species (Figure 9). After one cycle, the diffraction signals due to CaTiO_3_ and CaO species should have appeared at the end of regeneration. The diffraction signals assigned to CaCO_3_ species was due to the reaction of some CaO with CO_2_ as the experiment finished. The same situation occurred after 10 cycles. The SEM images of CaTi-3 showed more aggregated and fused morphology with increasing repetition times (Figure 10).

The different types of behavior of the CaTi-3 sorbent cyclic experiment under N_2_ flowing at low temperatures (700 °C for 30 min, mild condition) or under CO_2_ flowing at high temperatures (920 °C for 15 min, severe condition) may be related to calcium formation on the support surface based on the XRD results of sorbents collected after cyclic experiment (Figure 11). Under mild regeneration conditions, the cyclic experiment process mainly involves Ca(OH)_2_ and CaCO_3_ species transformation. Ca(OH)_2_ is under the molten state (melting point ~512 °C), while CaCO_3_ is in the solid state (melting point ~825 °C) during carbonation and regeneration. During carbonation, the ions diffuse in molten Ca(OH)_2_ layers more efficiently; thus, almost all calcium species can be utilized. When CaCO_3_ forms, it will precipitate due to its larger density (2.71 cm^3^/g) than Ca(OH)_2_ (2.21 cm^3^/g). During regeneration, the molten Ca(OH)_2_ layer permits the easy regeneration of CaCO_3_ located in the interior position. Therefore, nearly no capture capacity decay occurs. Under severe regeneration conditions, the cyclic experiment process mainly involves CaO and CaCO_3_ species transformation. CaCO_3_ is in the solid state during carbonation and under the molten state during regeneration; furthermore, CaO is in the solid state, and sintering cannot occur during carbonation and regeneration (melting point ~2613 °C, Tammann temperature ~1154 °C) [29]. During carbonation, external CaO can react with CO_2_ to form CaCO_3_, whereas CO_3_^2−^ and O^2−^ exchange occurs at the internal CaCO_3_/CaO interface [4]. During regeneration, CaO forms and precipitates in molten CaCO_3_ (density of CaO is 3.34 cm^3^/g). The microstructure stacked by these aggregated precipitates will affect subsequent cycle carbonation reaction performance. More precipitants with densely stacked structures reduce the carbonation efficiency.

## 4. Conclusions

We used a versatile aerosol-assisted self-assembly method to synthesize homogeneous CO_2_ adsorbent with calcium and titanium. The calcium titanium oxide species (CaTiO_3_) dispersed active species (CaO) well and mitigated the sintering effect during high-temperature multicycle carbonation and decarbonation. CO_2_-capture capacity increased with increased mole ratios of calcium and titanium. With a ratio equal to 3, the CO_2_-capture capacity became as large as 7 mmol/g with excellent reaction kinetics. Even after 20 cycles under mild regeneration conditions, the CO_2_-capture performance of CaTiO_3_-decorated CaO-based adsorbent was nearly retained with morphologies and structures unchanged. Moreover, after 10 cycles under severe regeneration conditions, the capture capacity still retained nearly 70% of the first cycle.

## Figures and Tables

**Figure 1 nanomaterials-11-03188-f001:**
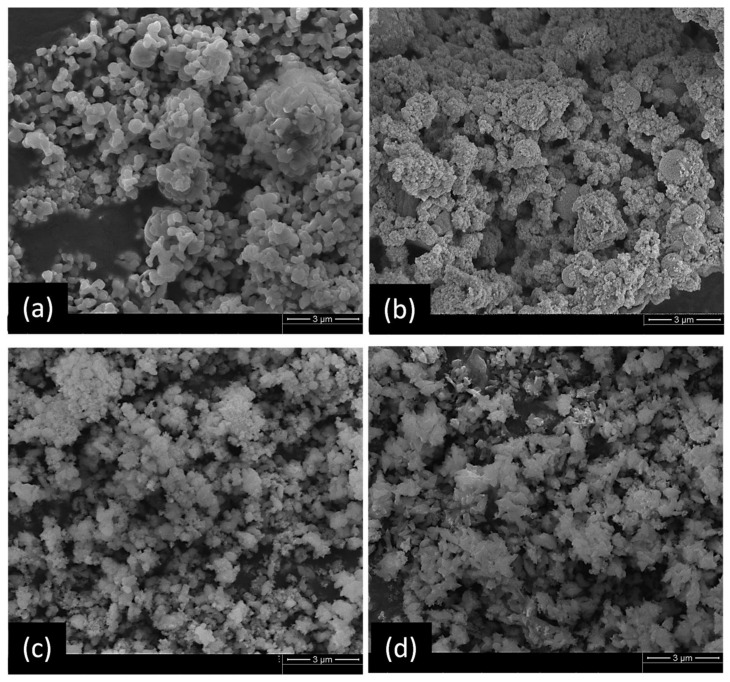
SEM images of as-prepared calcium-based sorbents: (**a**) CaTi-1.2, (**b**) CaTi-2, (**c**) CaTi-3, (**d**) CaTi-5.

**Figure 2 nanomaterials-11-03188-f002:**
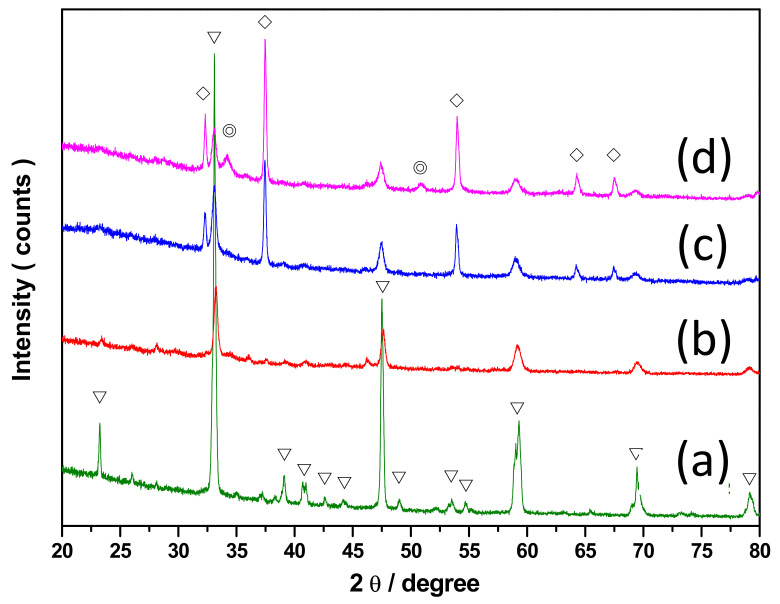
XRD result of as-prepared calcium-based sorbents: (**a**) CaTi-1.2, (**b**) CaTi-2, (**c**) CaTi-3, (**d**) CaTi-5. The inverted triangle symbol, diamond symbol, and circle symbol represent CaTiO_3_, CaO, and Ca(OH)_2_, respectively.

**Figure 3 nanomaterials-11-03188-f003:**
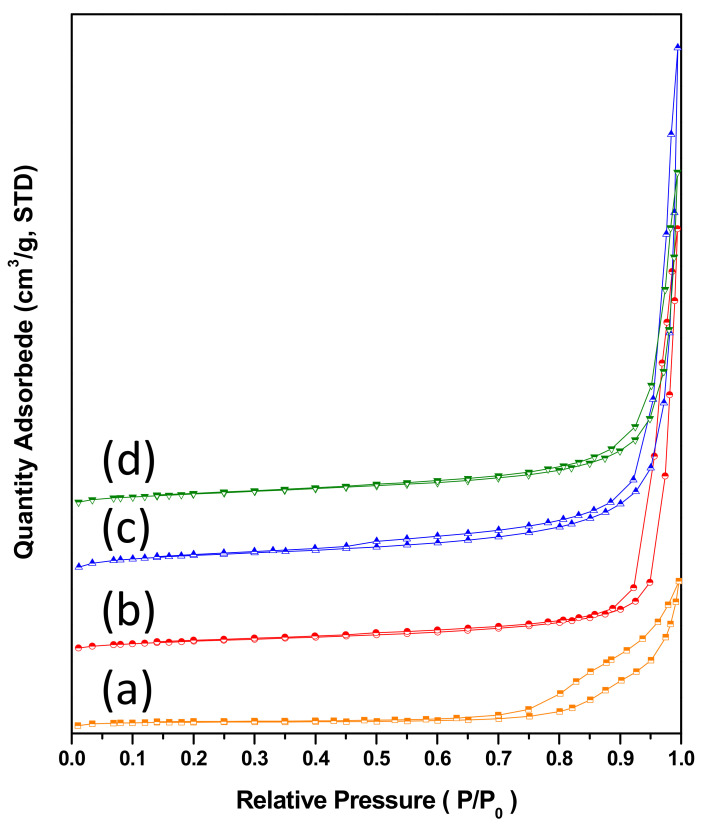
Nitrogen adsorption/desorption isotherm of as-prepared calcium-based sorbents: (**a**) CaTi-1.2, (**b**) CaTi-2, (**c**) CaTi-3, (**d**) CaTi-5.

**Figure 4 nanomaterials-11-03188-f004:**
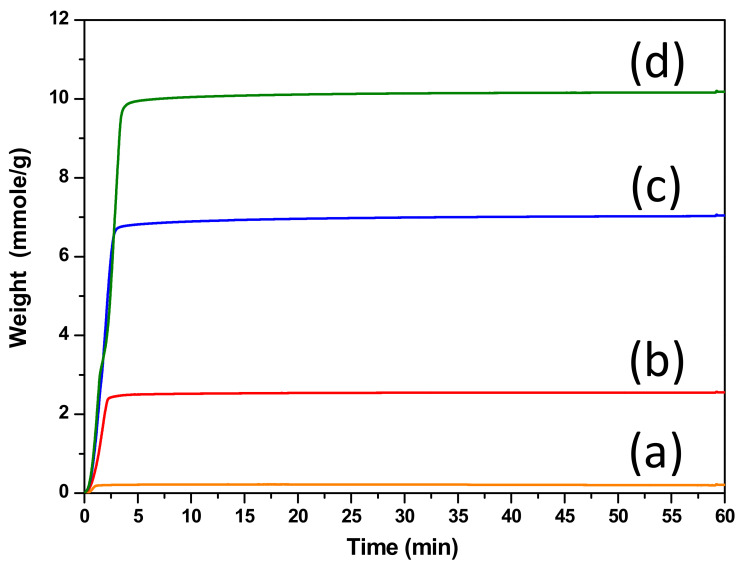
CO_2_ capture performance of calcium-based sorbents: (**a**) CaTi-1.2, (**b**) CaTi-2, (**c**) CaTi-3, (**d**) CaTi-5.

**Figure 5 nanomaterials-11-03188-f005:**
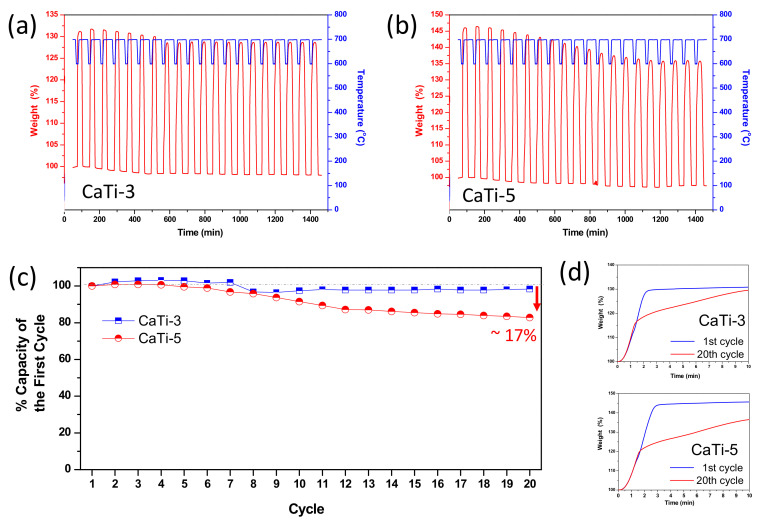
Twenty cyclic repeating experiment results of CaTi-3 (**a**) and CaTi-5 (**b**); percentage of capture capacity of each cycle to the first cycle (**c**), and the carbonation performance of the first and the twentieth experiment of CaTi-3 (up) and CaTi-5 (down) (**d**).

**Figure 6 nanomaterials-11-03188-f006:**
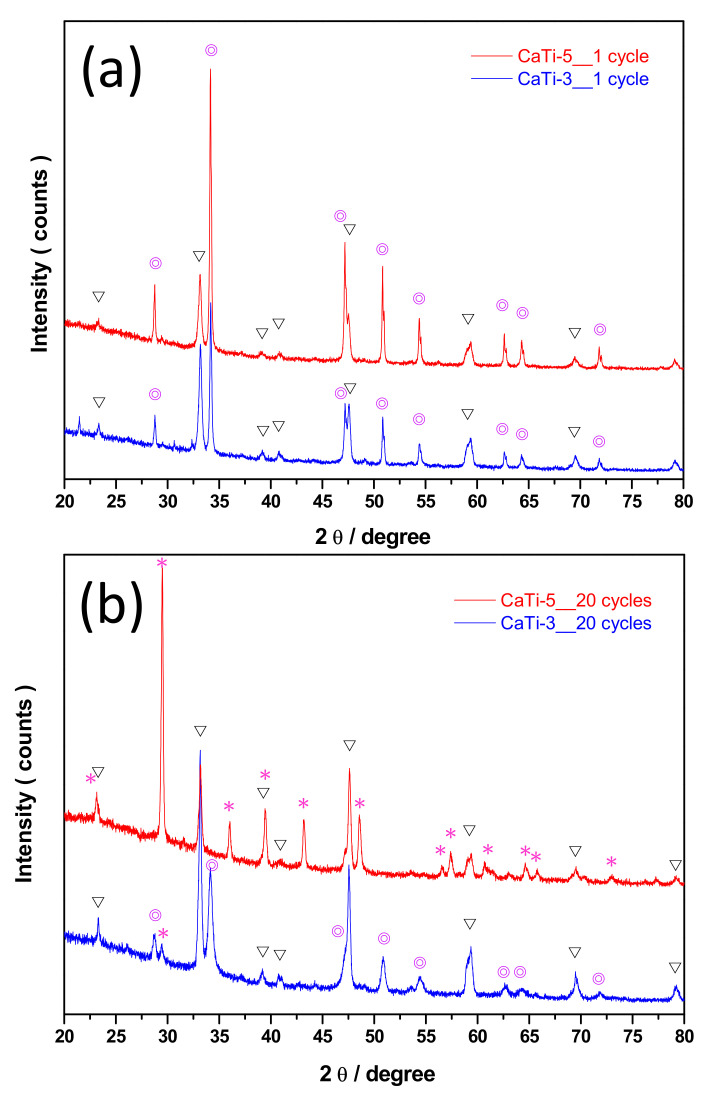
XRD results of (**a**) CaTi-3 and (**b**) CaTi-5 collected from after 1 cycle and 20 cycle experiments, where the invert triangle symbol represents CaTiO_3_, the circle symbol represents Ca(OH)_2_, and the star symbol represents CaCO_3_.

**Figure 7 nanomaterials-11-03188-f007:**
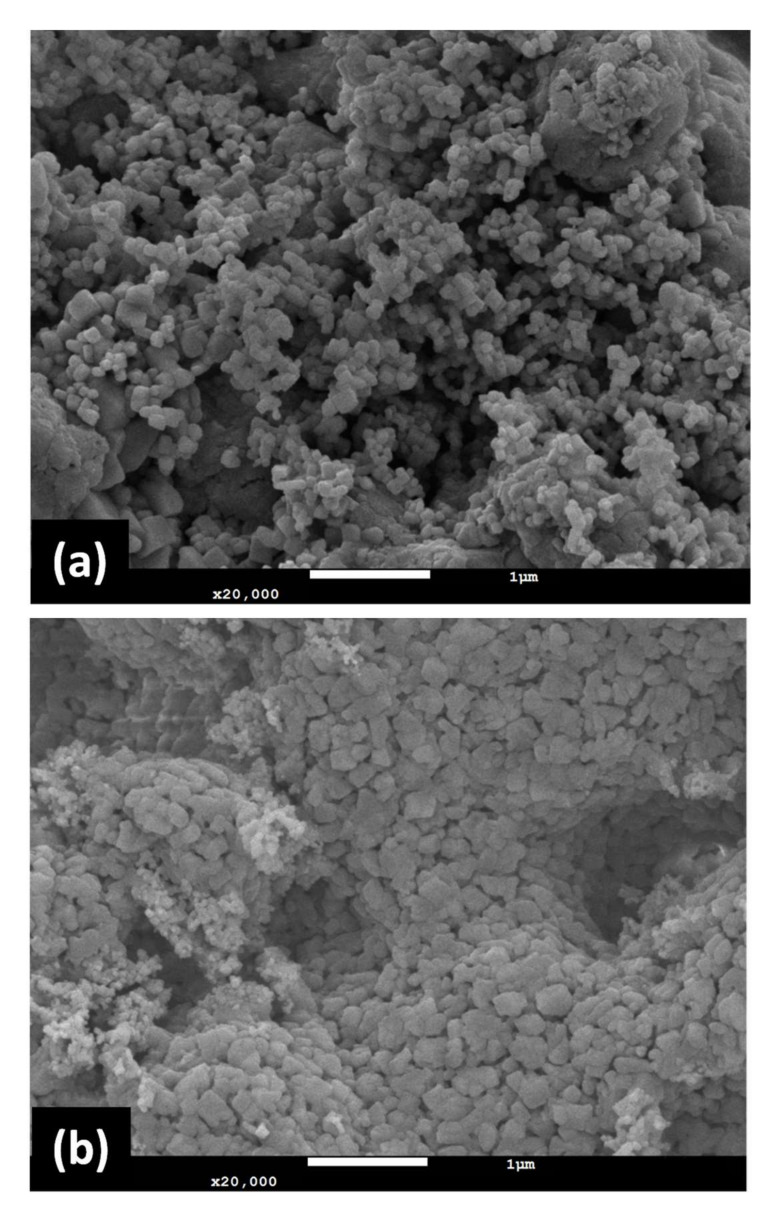
SEM images of CaTi-3 (**a**) and CaTi-5 (**b**) after 20 cycle experiment.

**Figure 8 nanomaterials-11-03188-f008:**
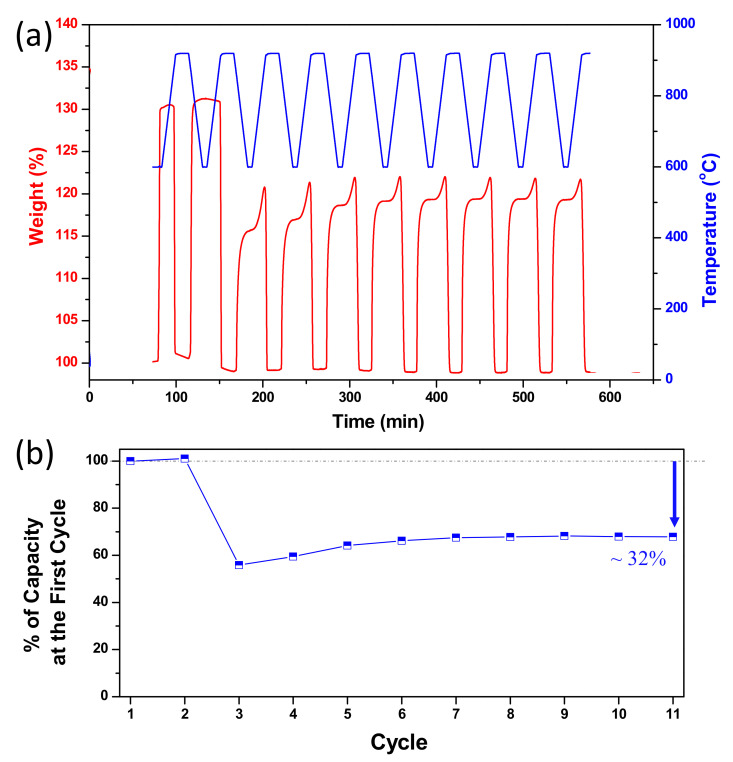
Ten cyclic repeating experiment results of CaTi-3 (**a**), and the percentage of capture capacity of each cycle to the first cycle (**b**).

**Figure 9 nanomaterials-11-03188-f009:**
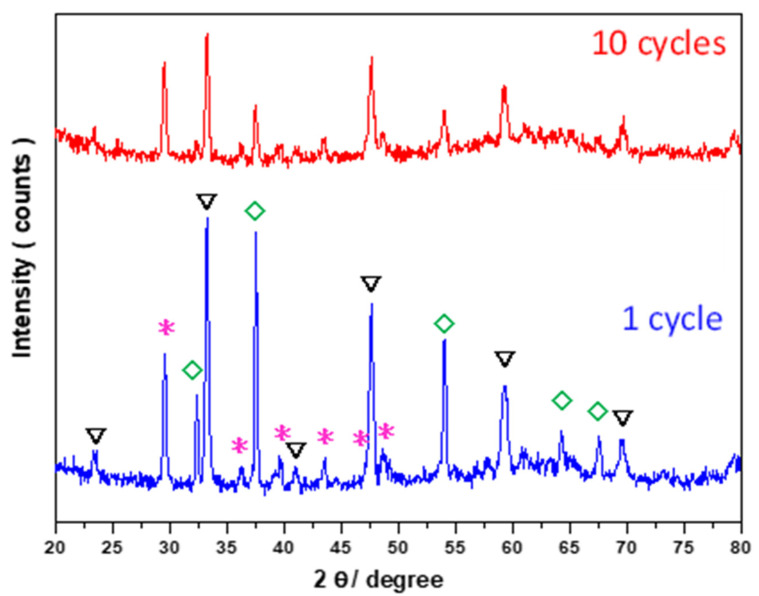
XRD results of CaTi-3 collected from after 1 cycle and 10 cycles experiment. Where the invert triangle symbol represents for CaTiO_3_, the diamond symbol represents for CaO, the star symbol represents for CaCO_3_.

**Figure 10 nanomaterials-11-03188-f010:**
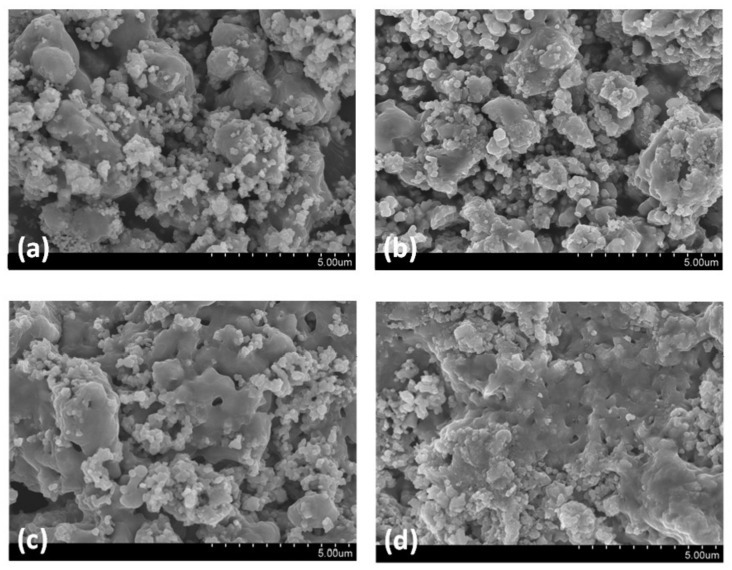
SEM images of CaTi-3 after (**a**) 1 cycle, (**b**) 2 cycle, (**c**) 3 cycle, and (**d**) 20 cycle experiments.

**Figure 11 nanomaterials-11-03188-f011:**
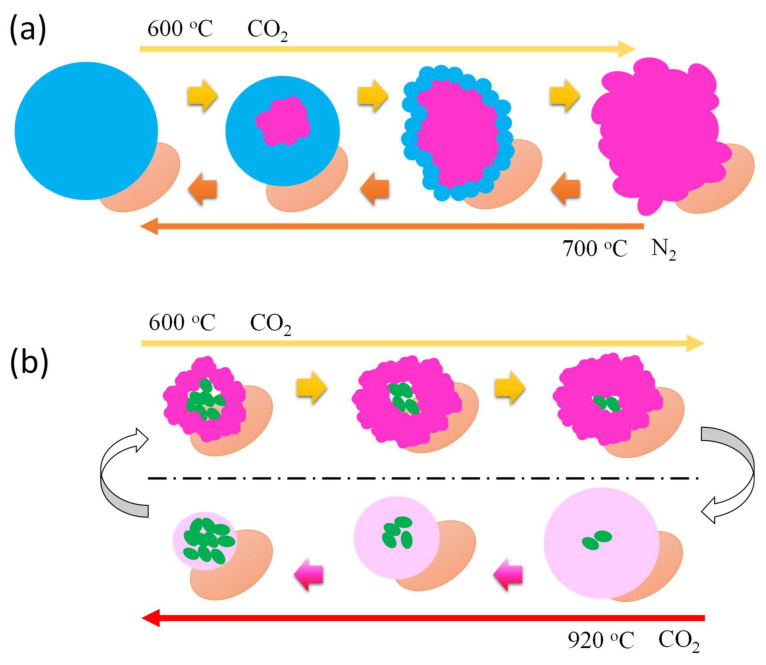
Schematic diagram of CaTi-3 under (**a**) mild regeneration conditions, and (**b**) severe regeneration of multicycle experiment.

**Table 1 nanomaterials-11-03188-t001:** Surface area and CO_2_ capture capacity of calcium-based sorbents.

Sorbent	Surface Area (m^2^/g)	CO_2_ Capture Capacity (mmole CO_2_/g Sorbent)
CaTi-1.2	5	0.2
CaTi-2	10	2.6
CaTi-3	17	7.0
CaTi-5	12	10.1

## Data Availability

The data presented in this study are available on request from the corresponding authors.

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
