# Peer review of "Multicycle Performance of CaTiO_3_ Decorated CaO-Based CO_2_ Adsorbent Prepared by a Versatile Aerosol Assisted Self-Assembly Method"

_nanomaterials, 2021, doi:10.3390/nano11123188_

Round 1
Reviewer 1 Report
The authors proposed a CaTiO3-decorated CaO-based material for the CO2 capture in the calcium looping process. the work is well written and I have only some minor revisions/comments:
Introduction: the authors cited different papers, which tried to improve sorbent performance, limiting the decay due to the sintering, but without describing the main results obtained by those authors. In other words, the introduction should be improved indicating the main results obtained by other authors.
Scope of the work: the authors should be better emphasize the scope of the work, because it is my opinion that is quite vague.
Page 6 line 123: the term ‘adsorption’ is written twice.
Author Response
Reply to Reviewers’ Comments and Suggestions
Comments (1)
Introduction: the authors cited different papers, which tried to improve sorbent performance, limiting the decay due to the sintering, but without describing the main results obtained by those authors. In other words, the introduction should be improved indicating the main results obtained by other authors.
Reply:
Thank you for your comment. Yu et al. used precipitation and deposition method to prepare Ca/Al/Ti sorbent. The capture capacity reduced less than 5% after 10 cycles capture-regeneration experiment as small amount of titania introduced. Wu et al. prepared CaTiO3/Nano-CaO by hydrolysis method. The authors found that high melting point of CaTiO3 appearance can improve cyclic stability of CaO-based adsorbent. The sorbent almost retain its capacity after 40 repeated use.
The description has adding to the manuscript.
Comments (2)
Scope of the work: the authors should be better emphasize the scope of the work, because it is my opinion that is quite vague.
Reply:
Thank you for your comment. Our scope is to prepare well sintering-resisted CaO-based CO2 adsorbent through easily scale up process. The description has adding to the manuscript.
Comments (3)
Page 6 line 123: the term ‘adsorption’ is written twice.
Reply:
Thank you for your comment. The problem has been corrected.
Reviewer 2 Report
The comments are attached.

Reviewer 3 Report
Review report attached as pdf file.

Author Response
Reply to Reviewers’ Comments and Suggestions
Specific comments (1)
Although not clearly stated in the text, it can be inferred from the SEM pictures that the sorbents were synthesized in the form of fine particles (average size < 50 micron). Even though such fine particles can be easily used and tested for characterization in static analysis systems, such as those used in this work, their applicability in actual dynamic systems even at lab-scale, such as fluidized bed reactors (which are typically used to perform calcium looping processes at large-scale), is very challenging.
In this framework, different solutions have been proposed to allow the use of such fine particles in real calcium looping processes. One possibility can be represented by the use of sound-assisted fluidization, as proposed in [a]. Another possibility is the modification of the CO2 sorbents to enhance their capture capacity and to increase their mechanical strength via: i) the preparation of granules with larger size and sufficient mechanical stability to withstand a fluidized bed process, as proposed in [b]; ii) the mixing of the sorbent particle with silica nano-powder, as proposed in [c,d].
These points should be discussed in the manuscript since the handling/processing of the sorbent material is a key issue in real CCS applications. The Authors should consider the reported works.
- Chem. Eng. J. Vol. 392, 2020, pp. 123658. https://doi.org/10.1016/j.cej.2019.123658.
- Renew. Energy Vol. 157, 2020, pp. 769–781. https://doi.org/10.1016/j.renene.2020.05.048
- Phys. Chem. Chem. Phys. Vol. 13, 2011, pp. 14906–14909. https://doi.org/10.1039/c1cp21939a
- Powder Technol. Vol. 249, 2013, pp. 443–455. https://doi.org/10.1016/j.powtec.2013.09.002
Reply:
Thank you for your comment. These information have been considered.
This strategy involves adding another metal (Zr, Al, Mg, Si, etc.) precursor during CaO-based adsorbent preparation, and then high-thermal-stability additives form independently (e.g., MgO, Al2O3, SiO2), or react with partial calcium (e.g., Ca9Al6O18, CaZrO3) as the sorbent is obtained, or introducing into high-thermal-stability porous silica solid (e.g., SBA-15, KIT-6) via impregnation process. The granules form sorbents for enhancing mechanical stability or acoustic sound-assisted technique are also considered to reduce sorbent deactivation after several repeated use.
The description have been adding in the introduction.
Specific comments (2)
Regarding the CO2 capture experiments, why did the Authors perform the capture stage (i.e. the carbonation stage) under pure CO2 instead of using a flue gas-like CO2 partial pressure/CO2 concentration (i.e. lower than 20%vol.)? The CO2-capture capacity is affected by the CO2 partial pressure and, therefore, the obtained values are overestimated with respect to those obtainable under typical flue gas conditions.
Reply:
Thank you for your comment. The flue gas contains usually lower than 20%vol. CO2. Therefore the capture capacity will be lower than under 100%vol. Higher capture capacity represents more CaO changes to CaCO3. However, because of different molar volume of CaO (16.7 cm3/mol) and CaCO3 (36.9 cm3/mol), the microstructure will be destroyed after several capture-regeneration looping due to frequently molar volume change. This phenomena reduces capture performance. The capture capacity should be lower under flue gas-like CO2 partial pressure, thus the microstructure destruction phenomena will be mitigated. We expect that the sorbent in this work will show much better cyclic stability in real application.
Specific comments (3)
Several grammatical errors/typos can be found throughout the text. Therefore, the manuscript should be carefully revised also from this point of view3
Reply:
Thank you for your comment. The grammatical errors in manuscript has been corrected by KGSupport institute for English language editing.
Round 2
Reviewer 3 Report
The manuscript has been revised/improved according to the Reviewer's comments/suggestion. Therefore, it can be now accepted for publication.